# Characterization of Lipopolysaccharide Effects on LRRK2 Signaling in RAW Macrophages

**DOI:** 10.3390/ijms24021644

**Published:** 2023-01-13

**Authors:** Asmaa Oun, Emmy Hoeksema, Ahmed Soliman, Famke Brouwer, Fabiola García-Reyes, Henderikus Pots, Marina Trombetta-Lima, Arjan Kortholt, Amalia M. Dolga

**Affiliations:** 1Department of Molecular Pharmacology, Faculty of Science and Engineering, Groningen Research Institute of Pharmacy (GRIP), University of Groningen, 9713 AV Groningen, The Netherlands; 2Department of Cell Biochemistry, Groningen Institute of Biomolecular Sciences & Biotechnology (GBB), University of Groningen, 9747 AG Groningen, The Netherlands; 3Department of Biotechnology, Institute of Graduate Studies and Research, Alexandria University, Alexandria 21526, Egypt; 4YETEM-Innovative Technologies Application and Research Centre, Suleyman Demirel University, 32260 Isparta, Turkey

**Keywords:** LRRK2, inflammation, LPS, cytokines

## Abstract

Dysfunction of the immune system and mitochondrial metabolism has been associated with Parkinson’s disease (PD) pathology. Mutations and increased kinase activity of leucine-rich repeat kinase 2 (LRRK2) are linked to both idiopathic and familial PD. However, the function of LRRK2 in the immune cells under inflammatory conditions is contradictory. Our results showed that lipopolysaccharide (LPS) stimulation increased the kinase activity of LRRK2 in parental RAW 264.7 (WT) cells. In addition to this, LRRK2 deletion in LRRK2 KO RAW 264.7 (KO) cells altered cell morphology following LPS stimulation compared to the WT cells, as shown by an increase in the cell impedance as observed by the xCELLigence measurements. LPS stimulation caused an increase in the cellular reactive oxygen species (ROS) levels in both WT and KO cells. However, WT cells displayed a higher ROS level compared to the KO cells. Moreover, *LRRK2* deletion led to a reduction in interleukin-6 (IL-6) inflammatory cytokine and cyclooxygenase-2 (COX-2) expression and an increase in lactate production after LPS stimulation compared to the WT cells. These data illustrate that LRRK2 has an effect on inflammatory processes in RAW macrophages upon LPS stimulation.

## 1. Introduction

Parkinson’s disease (PD) is the second most common neurodegenerative disorder affecting the elderly population. The incidence of the disease increases with age and about 1% of the population over 60 years are affected [1]. Degeneration of dopaminergic neurons in the substantia nigra pars compacta (SNpc) is the hallmark of PD and the reason underlying the motor symptoms of the disease [2]. PD has a multifactorial etiology where both genetic and environmental factors are involved in disease pathogenesis [3]. Familial PD accounts for a small percentage of cases, while the rest are sporadic with unknown etiology [4]. Among the pathogenic gene variants of PD, leucine-rich repeat kinase 2 (*LRRK2*) represents at least 5% of familial and 1% of sporadic cases [5]. LRRK2 is a large multidomain protein with both kinase and GTPase functions [6,7]. The increased kinase activity of LRRK2 has been reported in both familial and sporadic PD cases [8,9], indicating that studying LRRK2 can provide insights into common mechanisms in both forms of the disease.

LRRK2 has been implicated in several biological processes that are linked to PD pathogenesis, including mitochondrial function, inflammation and immunity, autophagy, cell signaling and neurite growth and differentiation [10,11]. Moreover, more evidence has recently become available supporting the role of LRRK2 in immune function and its implication in PD [12]. Neuroinflammation represents a hallmark of PD [13], and LRRK2 is highly expressed in both central and peripheral immune cells, particularly microglia, macrophages, B-lymphocytes and dendritic cells [14,15,16]. In addition, LRRK2 is upregulated in the immune cells of PD patients compared to healthy controls [17,18], which is associated with the enhanced kinase activity of LRRK2. Noticeably, LRRK2 is associated with risk for a number of immune disorders including leprosy [19] and Crohn’s disease [20]. In total, these data indicate that LRRK2 plays an important role in immune function. However, the exact mechanism of LRRK2 in the immune process is not defined yet.

Lipopolysaccharide (LPS) is a toll-like receptor (TLR4) agonist [15] that is commonly used as an inflammatory stimulus to mimic neuroinflammation in PD [21]. LPS injection in animal PD models was able to induce microglial activation and loss of dopaminergic neurons in the SN mimicking the hallmarks of PD pathology [22,23,24,25,26]. Additionally, LRRK2 expression is increased by inflammatory stimuli such as interferon-gamma (INF-γ) [14,27,28] and LPS [15,28,29,30] in different immune cell models for PD. In addition, LPS was found to increase the kinase activity of LRRK2 [29,31,32,33,34]. Although a consistent increase in inflammatory cytokines production occurs upon LPS stimulation in immune cells, conflicting results regarding the regulatory function of LRRK2 on cytokine production have been reported in immune cells [15,29,31,35,36]. We, therefore, decided to use LPS-activated macrophages to characterize the role of LRRK2 on the inflammatory process in peripheral immune cells.

Murine LRRK2 parental RAW 264.7 (WT) cells and LRRK2 KO RAW 264.7 (KO) cells were used to study the effects of LRRK2 upon LPS stimulation. LPS stimulation increased LRRK2-dependent substrate phosphorylation. In addition, *LRRK2* deletion led to an increase in cell impedance after LPS stimulation, indicating an alteration in cell morphology. Moreover, LRRK2 deletion resulted in a reduction in cellular reactive oxygen species (ROS), interleukin-6 (IL-6) and cyclooxygenase-2 (COX-2) levels and an increase in lactate production. These results indicate that LRRK2 plays a role in immune cell adaptation to inflammatory stimuli.

## 2. Results

### 2.1. LPS Increases the Kinase Activity of LRRK2

It was shown previously that LRRK2 expression can be induced using different TLR agonists, with the highest expression level using LPS (TLR4 agonist) [15]. Increased expression of LRRK2 has been reported in different immune cell models [15,28,29,30]. Our results also show that murine LRRK2 parental RAW 264.7 (WT) cells show a slight, but not significant, increase in LRRK2 expression after LPS stimulation (Appendix A).

To confirm the increased kinase activity of LRRK2 in our model, we performed immunoblotting for Rab8 and phospho Rab8. The phospho Rab8 antibody cross-reacts with a subset of LRRK2 phosphorylated Rabs (Rab3A, Rab8, Rab10, Rab35 and Rab43) and provides a good indication of LRRK2 effects on Rab phosphorylation in general. The WT, KO and kinase-inhibited WT cells were stimulated with LPS (250 ng/mL) for 6 and 24 h. The ratio of phosphorylated Rab8 to total Rab8 protein was calculated. Our results show that the level of phosphorylated Rab8 was generally low in the KO and kinase-inhibited WT cells in the presence or absence of LPS stimulation compared to the WT (at 6 h, the signal for phosphorylated Rab8 is almost undetected in the KO and kinase-inhibited cells, and after 24 h, only a faint band is detected; that is why we focused our analysis on the WT cells) (Figure 1 A,C). LPS stimulation provoked the kinase function of LRRK2, and it was significantly higher in the WT cells after 24 h of LPS stimulation compared to the untreated cells (Figure 1C,D). These results confirm that LPS stimulation can increase the kinase function of LRRK2.

### 2.2. LRRK2 Deletion Changes the Cell Morphology after LPS Stimulation

To investigate the effects of LRRK2 on the macrophages’ behavior upon LPS stimulation, we first characterized morphological changes in the WT and KO cells in real-time after the application of increasing concentrations of LPS (0–1µg/mL). Using xCELLigence real-time impedance measurements over 24 h, we showed that the WT and KO cells have a similar cell index under unstimulated conditions indicating similar morphology and growth rate (Appendix A). Following the application of increasing concentrations of LPS, both WT and KO cells showed an increase in cell area indicated by an increase in the normalized cell index compared to unstimulated cells. However, the increase in the cell index was higher in stimulated KO cells compared to stimulated WT cells in all concentrations of LPS used (Figure 2 and Appendix A). Microscopy analyses showed that the change in the normalized cell index was not due to a change in cell division after LPS treatment (Appendix A). The difference between stimulated WT and KO was the highest and showed statistical significance using 250 ng/mL LPS (Figure 2), and we continued our assays with this LPS concentration. These data suggest that LRRK2 regulates morphological changes in RAW macrophages after LPS stimulation.

### 2.3. LRRK2 Mediates LPS-Stimulated ROS Production

Previous studies have shown that stimulation of immune cells with LPS increased cellular ROS levels [37,38,39]. To investigate the role of LRRK2 on LPS-induced ROS production, flow cytometry measurement of 2′,7′-dichlorodihydrofluorescein diacetate (DCFDA) was performed on WT, KO and kinase-inhibited WT cells after 24 h of LPS stimulation. DCFDA is a cell-permeable non-fluorescent dye that is transformed to oxidation-sensitive dye by intracellular esterases, dichlorodihydrofluorescein (DCF), that can be measured by flow cytometry [40]. Our results showed that cellular ROS levels are similarly low in the unstimulated WT and KO cells (Figure 3A,B), while inhibition of the kinase function of LRRK2 resulted in a slight increase in the ROS level in unstimulated cells (Figure 3C). LPS stimulation results in a significant increase in the ROS levels of the WT and KO cells, yet the increase in the ROS levels in the kinase-inhibited WT was not significant compared to the unstimulated cells (Appendix A). The mean DCF fluorescence in stimulated KO cells was significantly lower than in stimulated WT cells (Figure 3B), while the stimulated kinase-inhibited WT cells showed a similar, but not significant, trend towards ROS reduction, suggesting that LRRK2 mediates LPS-stimulated ROS production.

### 2.4. Inflammatory Cytokine Production in Stimulated RAW Macrophages

LPS stimulation of immune cells increases inflammatory cytokine production. However, the effect of LRRK2 in regulating inflammatory cytokines upon LPS stimulation in immune cells is inconsistent [15,29,31,35,36]. LPS is known to increase the level of the proinflammatory cytokines IL-6, tumor necrosis factor alpha (TNF-α) and interleukin-1 beta (IL-1β) in immune cells through its binding to TLR4 and activation of nuclear factor-kappa B (NF-κB) which in turn induces the transcription of proinflammatory cytokines [41,42]. The results from reverse transcription quantitative real-time PCR (qPCR) and enzyme-linked immunosorbent assay (ELISA) measurements showed that IL-6 levels in unstimulated cells was very low and increased after LPS stimulation (Appendix A). The comparison between WT and KO cells revealed no difference in IL-6 expression in unstimulated cells. However, after LPS stimulation, IL-6 was significantly reduced in the KO cells compared to the WT (Figure 4). These results were consistent using both qPCR (Figure 4A–C) and ELISA (Figure 4D,E) measurements. Kinase inhibition of LRRK2 resulted in a similar reduction in IL-6 level that was significant after 6 h of LPS stimulation using qPCR and 24 h using ELISA (Appendix A). The results suggest that LRRK2 and its kinase function might influence IL-6 production.

The TNF-α level was increased after LPS stimulation compared to unstimulated cells (Appendix A). However, no differences in TNF-α between the WT and KO cells were detected (Figure 5). Only the KO cells showed an increase in TNF-α using qPCR after 6 h of LPS stimulation (Figure 5E). Our results indicate no effect from LRRK2 on TNF-α production.

The IL-1β expression did not change upon LPS stimulation in the WT and KO cells while a reduction in the basal expression of IL-1β was detected in the unstimulated KO cells compared to the WT (Appendix A).

### 2.5. LRRK2 Deletion Reduces COX-2 Expression

Previous studies reported increased expression of COX-2 in neurons [43] and microglia [44] of the SN in postmortem PD patients and PD mice models [43]. Selective inhibition of COX-2 reduced microglial activation and degeneration of dopaminergic neurons in different PD models [45,46]. These data support the contribution of COX-2 in PD pathogenesis. Deletion of *LRRK2* in primary microglia, BV2 cells [35] and PD patients’ fibroblasts [47] resulted in reduced expression of COX-2 upon LPS stimulation. However, the link between LRRK2 and COX-2 has not been studied before in peripheral immune cells. Here, we measured the level of COX-2 expression following LPS stimulation using immunoblotting and flow cytometry. Our immunoblotting results showed that a COX-2 signal cannot be detected without LPS stimulation, and quantification of the signal after 6 and 24 h of LPS stimulation revealed a reduction in COX-2 expression in the KO cells compared to the WT (Figure 6D–G). For more quantitative measurement of COX-2 expression, flow cytometry was used. COX-2 expression was significantly lower in the KO cells than the WT under basal conditions (Figure 6A,B). Additionally, COX-2 expression after 24 h of LPS stimulation was significantly reduced in the KO cells (Figure 6A,C), similar to the immunoblotting data. These data suggest that LRRK2 acts upstream of COX-2 transcription.

### 2.6. LRRK2 Influences Glycolysis following LPS Stimulation

Stimulation of macrophages leads to metabolic reprogramming and increased glycolysis [48]. To investigate the role of LRRK2 in the glycolysis process, we measured the lactate level in the cell supernatant of the WT, KO and kinase-inhibited cells after LPS stimulation. Lactate production was increased after LPS stimulation. There was no significant change in the lactate production in unstimulated cells (Figure 7A). LPS-stimulated KO cells showed a significant increase in lactate production compared to the WT cells after 24 h (Figure 7). Our results verify previous studies showing a link between LRRK2 and glycolysis.

## 3. Discussion

Only recently, LRRK2 function to regulate the immune system both in the central nervous system and the periphery was demonstrated [49] with evidence of the interplay between the central and peripheral immune systems in PD pathogenesis [50,51]. Therefore, more knowledge about LRRK2 effects under inflammatory conditions in the peripheral immune system is demanded.

Increased LRRK2 phosphorylation after LPS stimulation has been previously reported in different immune cell models using pS935 antibody [29,31,32,33,34,52]. Our data showed that LPS stimulation of the WT cells increased the kinase function of LRRK2 measured by the immune blotting of the LRRK2 substrate phospho Rab8. Phosphorylation of Rab8 was reported after intracellular pathogen infection and application of endolysosomal membranolytic agent (L-leucyl-L-leucine methyl ester (LLOMe)) in RAW 264.7 cells [53]. Following TLR4 activation by LPS, Rab8 was involved in inflammatory signaling through recruitment of phosphatidylinositol 3-kinase γ (PI3Kγ) for Akt-dependent signaling, indicating that LRRK2-mediated phosphorylation could be involved in inflammation via Rab8 [54]. In contrast to our findings, a recent study showed that Rab8 phosphorylation was not altered upon LPS stimulation cells in LRRK2 WT RAW 264.7 cells [52]. This could be due to using a lower concentration of LPS (100 ng/mL) and different duration. The low signal of phospho Rab8 in the KO cells and the reduction in pRab8 signal in kinase-inhibited WT cells confirm that Rab8 is a bona fide substrate of LRRK2.

Oxidative stress has been linked to PD pathogenesis as increased ROS production can lead to the degeneration of dopaminergic neurons [55]. Our results showed that LPS stimulation increases ROS production in RAW macrophages. In addition to this, deletion or kinase inhibition of LRRK2 reduced ROS production following LPS stimulation. These results align with previous reports linking LRRK2 and its kinase function to increased ROS production. Reduction in the ROS level in KO LRRK2 RAW macrophages following induction of oxidative stress using manganese [56] and zymosan [14] compared to LRRK2 WT RAW was previously reported. Moreover, kinase inhibition of LRRK2 lowered the ROS level in LRRK2 WT RAW 264.7 cells and human microglia HMC3 cells following manganese-induced oxidative stress [56]. In addition to the inhibition of LRRK2 kinase function using ATP-competitive LRRK2 kinase inhibitors, the kinase activity of LRRK2 can be reduced using constrained peptides that inhibit the dimerization of LRRK2 and, hence, its kinase function. The use of constrained peptides successfully lowered the ROS level after zymosan stimulation of LRRK2 WT RAW cells [57]. Collectively, these findings confirm the involvement of LRRK2 and its kinase function in oxidative stress. One study reported an increase in the accumulation of 4-hydroxynonenal (4-HNE), indicating an increase in oxidative stress after 12 h of LRRK2 kinase inhibition in SH-SY5Y cells [58]. This is in agreement with our results that kinase inhibition of LRRK2 for 24 h showed a slight increase in the ROS level in WT cells, suggesting that long incubation with LRRK2 kinase inhibitors might increase the ROS production in a time-dependent manner.

Regulation of inflammatory cytokine production is an important function of innate immune cells in PD pathogenesis. Higher levels of inflammatory cytokines have been reported in PD patients [59,60,61]. Levels of serum inflammatory cytokines have been studied in PD patients carrying *LRRK2* mutations [62,63], emphasizing the important role of inflammatory cytokines.

In this study, we demonstrated a reduction in IL-6 production in the KO and kinase-inhibited WT cells following LPS stimulation at the transcriptional and translational levels compared to the WT. Previous reports exhibited contradicting findings regarding the IL-6 level in different LRRK2 models. In line with our findings, IL-6 level was reduced in LPS-stimulated BV2 microglia [36] and induced pluripotent stem cell (iPSC)-derived microglia [64] in KO LRRK2 cells compared to the control. This reduction might be due to reduced phosphorylation of p38 mitogen-activated protein kinase (MAPK) involved in inflammatory cytokine production [36]. Other studies reported no change [15,31,52,65,66] or even an increase [67] in IL-6 level upon LPS stimulation in the LRRK2 KO cells compared to the WT. Discrepancies in IL-6 levels could be related to the immune cell model used or the duration and concentration LPS stimulation. In addition, IL-6 exhibits both pro- and anti-inflammatory properties [68], which means IL-6 is involved in complex cellular processes and further research is needed to investigate the relationship between LRRK2 and IL-6.

Our results showed no influence of LRRK2 on TNF-α and IL-1β upon LPS stimulation in RAW cells. In agreement with our data, no change in TNF-α [31,52,66] and IL-1β [31,65,66] was observed in immune cells lacking LRRK2 compared to WT cells after LPS stimulation. A reduction in TNF-α in BV2 cells [36] and iPSC-derived microglia [64] was reported in the LRRK2 KO cells after LPS stimulation. IL-1β was also reduced after LPS stimulation in microglial cells lacking LRRK2 [35,36] or upon LRRK2 kinase inhibition [35].

Previous studies supported the contribution of COX-2 in PD pathogenesis through the reduction in microglial activation and degeneration of dopaminergic neurons in PD models by selective inhibition of COX-2 [45,46]. A recent epidemiological study has proposed that the regular use of non-steroidal anti-inflammatory drugs (NSAIDs) that inhibit COX-2 could reduce the risk of PD in carriers of both pathogenic and risk variants of *LRRK2* [69], suggesting a link between LRRK2 and COX-2 in PD pathogenesis. Our data demonstrate that deletion of LRRK2 reduced the COX-2 protein level in the KO cells upon LPS stimulation compared to the WT. Our data are in agreement with a previous study showing that Lrrk2^-/-^ primary microglia show reduced expression of COX-2 on the protein levels upon LPS stimulation [35]. In the same study, the use of LRRK2 kinase inhibitor (LRRK2-IN-1) in LPS-stimulated BV2 cells lowered COX-2 expression at the RNA and protein levels. The decrease in COX-2 expression may be due to the increased phosphorylation and nuclear levels of the NF-κB inhibitory subunit resulting in a reduction in NF-κB target-gene transcription [35]. Another study revealed that the knockdown of LRRK2 in PD patients’ fibroblast resulted in a reduction in COX-2 RNA levels after LPS stimulation [47].

Enhanced glycolysis was found to attenuate PD symptoms in several disease models [70]. The association between LRRK2 and glycolysis has been described [71]. In fact, LRRK2 was found to contribute to metabolic reprogramming after stimulation in immune cells [72]. In this study, we quantified lactate released in the medium from the cells with and without LPS stimulation as a measure for glycolytic flux [73]. Our data displayed no change in lactate release in unstimulated WT and KO cells, while lactate production was increased in the KO cells after 24 h of LPS stimulation compared to WT cells. In line with our data, LRRK2^-/-^ mouse embryonic fibroblasts (MEFs) displayed similar glycolysis to LRRK2^+/+^ MEFs under basal conditions [74]. However, LRRK2 KO bone-marrow-derived macrophage (BMDMs) had reduced glycolysis compared to the LRRK2 WT cells in another study [75]. Following LPS stimulation, LRRK2 KO induced pluripotent stem-cells-derived microglia [64] and LRRK2 KO bone-marrow-derived macrophages [72] showed reduced glycolysis compared to WT LRRK2 cells. Discrepancies in the change in glycolysis could be attributed to the cell model used and the approach used to measure glycolysis. Glycolysis was examined in the aforementioned studies by measuring extracellular acidification rate (ECAR) using an XFe extracellular flux analyzer.

Altogether, our data suggest a role of LRRK2 in ROS, IL-6 and COX-2 production under inflammatory conditions in macrophages (Figure 8). In addition, LRRK2 contributes to glycolysis regulation in immune cells. These data could help identify targets to modulate neuroinflammation in PD through LRRK2 function. Contradictions in the field in relation to LRRK2 role in inflammation could be related to the model or cell type used, but also serve as an indication of the complex role of LRRK2. Therefore, comparing the effect of LRRK2 under stimulatory conditions in both central (microglia) and peripheral immune cells (macrophages and monocytes) could resolve the cell-type-related effects of LRRK2. Moreover, the use of human induced pluripotent stem-cell-derived immune cells to study the immune function of LRRK2 could bridge the gap between animal models and human physiology. In addition, the choice of inflammatory stimulus is important to investigate specific signaling pathways through which LRRK2 could modulate the inflammatory processes. Finally, comparing different gain-of-function mutations of LRRK2 besides the WT and KO models could provide more knowledge on the role of LRRK2 under the increased kinase activity reported in PD patients. Our data indicate a complex role of LRRK2 in inflammatory conditions in peripheral macrophages that could affect PD pathology.

## 4. Materials and Methods

### 4.1. Cell Culture

LRRK2 parental RAW 264.7 (ATCC^®^ SC-6003TM) and LRRK2 KO RAW 264.7 (ATCC^®^ SC-6004TM) were obtained from ATCC. The cells were cultivated in growth medium (Dulbecco’s Modified Eagle Medium (DMEM) (ATCC^®^ 30-2002™), supplemented with 10% fetal bovine serum (ATCC^®^ 30-2020™) and 1% penicillin-streptomycin (5000 U/mL) (Gibco™, 15070063)). The cells between passage 3–15 were used for the experiments. MycoAlert PLUS Mycoplasma Detection Kit (Lonza, Basel, Switzerland, LT07-710) was regularly used to confirm that the cells were mycoplasma-free.

### 4.2. Cell Treatment

LRRK2 parental and KO RAW 264.7 cells were treated with lipopolysaccharide (LPS) 250 ng/mL (from Escherichia coli O55:B5, Sigma, St. Louis, MI, USA, L2880) for 6 and 24 h (unless otherwise stated). To check the effect of the kinase function of LRRK2, the cells were pretreated with MLi2 1 µM (Tocris, Bristol, UK, 5756) for 90 min before LPS treatment. MLi2 was maintained in the medium during the treatment time.

### 4.3. Western Blotting

The cells were seeded in 6-well plates at a seeding density of 500,000 cells/well overnight before LPS or MLi2 pretreatment and LPS application. Following the incubation, the cells were washed with ice-cold 1XPBS and collected with the corresponding lysis buffer. For Rab8 and phospho Rab8 detection, the cells were lysed for 15 min at 4 °C in the lysis buffer (50 mM Tris-HCl; pH 7.5, 1% (*v*/*v*) Triton X-100, 1 mM EGTA, 1 mM Na3VO4, 50 mM NaF, 10 mM β-glycerophosphate, 5 mM sodium pyrophosphate, 270 mM sucrose, cOmpleteTM (EDTA-free Protease Inhibitor Cocktail (Roche, 11836170001)), 0.1 μg/mL Microcystin-LR (Enzo Life Sciences, ALX-350-012) and 0.5 mM DIFP (Sigma, Cat# D0879)). The cell lysates were centrifuged at 18,000× *g* for 20 min at 4 °C. For COX-2 detection, the cells were lysed in 1% SDS lysis buffer (10 mM Tris, 1 mM EDTA, 1% SDS; pH 8, 1 mM Na3VO4, 50 mM NaF, 10 mM β-glycerophosphate, 5 mM sodium pyrophosphate, 1X P2714 (Sigma (P2714-1BTL)). After that, the lysates were sonicated 2 times with 5 pulses at 40 Amp then centrifuged at 11,292× *g* for 15 min at 4 °C. The total protein content of the supernatants was determined using a Pierce™ BCA Protein Assay kit (Thermo Scientific, Waltham, MA, USA, #23225). For Rab8 and phospho Rab8, 60 µg of the total protein per sample was loaded on 12% SDS-PAGE gel, and for COX-2 and LRRK2, 30 µg of the total protein per sample was loaded on 10% SDS-PAGE gel. Following electrophoresis and transfer, the membranes were incubated with the corresponding primary antibodies overnight at 4 °C at 1:1000 dilution. The primary antibodies used are Rab8A (D22D8) XP^®^ (Cell Signaling, Danvers, MA, USA, 6975), anti-RAB8A (phospho T72) (Abcam, Cambridge, MA, USA, ab230260), COX-2 (D5H5) XP^®^ (Cell signaling, Danvers, MA, USA, 12282), recombinant anti-LRRK2 antibody [MJFF2 (c41-2)] (Abcam, ab133474) and GAPDH (14C10) (Cell signaling, Danvers, MA, USA, 2118S) as a loading control. After washing, the membranes were incubated with IRDye^®^ 800CW goat anti-rabbit IgG secondary antibody (LI-COR, 926-32211) for 2 h at room temperature and visualized using a LI-COR^®^ Odyssey^®^ Fc Dual-Mode Imaging System (LI-COR^®^). The images were analyzed using Image Studio™ software.

### 4.4. Real-Time Cell Impedance Measurements

Real-time measurements of morphological changes in LRRK2 parental and KO RAW 264.7 cells were performed using a label-free xCELLigence^®^ RTCA MP system (ACEA BIO, Westerburg BV, Leusden, The Netherlands) [76]. A total of 20,000 cells/well were seeded in a 96-well E-plate (Agilent, 5232368001) containing gold microelectrodes fused to the bottom surface of the well plate. The cell index, representing the impedance of the electron flow caused by cell attachment to the well and changes in cell morphology, was recorded every 30 min. The cells were seeded one day before treatment and the measurement of the cell index was performed 24–28 h following LPS application. The cell index was normalized to 1, before the addition of various compounds.

### 4.5. Flow Cytometry Measurement

LRRK2 parental and KO RAW 264.7 cells were seeded in a 24-well plate (80,000 cells/well) one day before LPS challenge. Flow cytometry measurements were performed via the CytoFLEXS benchtop flow cytometer (Beckman Coulter Life Sciences, Indianapolis, IN, USA). At least three independent experiments were performed with three technical replicates per condition. A total of 30,000 events were counted per technical replicate and data were analyzed with flowjo-V10 software.

#### 4.5.1. Reactive Oxygen Species (ROS) Levels

After 24 h of LPS treatment, the cells were collected by trypsinization and incubated with 2.5 µM of the cell-permeant 2’,7’-dichlorodihydrofluorescein diacetate (H2DCFDA) (Invitrogen™, Carlsbad, CA, USA, D399) in serum-free DMEM for 30 min. Fluorescence was detected using an FITC filter.

#### 4.5.2. Cyclooxygenase-2 (COX-2) Detection

COX-2 detection was performed following the manufacturer’s protocol. Briefly, the cells were dissociated from plates by trypsinization and fixed with 4% paraformaldehyde for 15 min at room temperature. Then, they washed with excess 1XPBS, permeabilized by adding ice-cold 100% methanol to a 90% final methanol concentration and stored at −20 °C. For immunostaining, the cells were washed in excess 1XPBS, resuspended in COX-2 primary antibody at 1:100 dilution in antibody dilution buffer (0.5% BSA in 1XPBS) and incubated for 1 h at room temperature. After washing, the cells were incubated in 1:1000 Alexa Fluor 488 and goat anti-rabbit IgG (Invitrogen™, Carlsbad, CA, USA, A11034) for 30 min at room temperature. After that, the cells were washed, resuspended in 1XPBS and analyzed by flow cytometer using the FITC filter.

### 4.6. RNA Extraction and Reverse Transcription Quantitative Real-Time PCR (RT-qPCR)

For RNA extraction, the cells were seeded in 12-well plates (260.000 cells/well) overnight before LPS stimulation. After the treatment, RNA was extracted using TRIzol™ reagent (Invitrogen™, 15596026) according to the manufacturer’s protocol. A total of 1μg RNA was transcribed into cDNA using reverse transcription by M-MLV Reverse Transcriptase (Promega, M1701). RT-qPCR was performed in the presence of FastStart Universal SYBR Green Master (Rox) (Sigma, 4913914001) using an Illumina Eco Real-Time PCR System (Westburg, Leusden, The Netherlands). PCR started with 10 min of initial denaturation at 95 °C followed by 45 cycles of denaturation at 95 °C for 30 s, annealing at 55 °C for 30 s and extension at 72 °C for 30 s with a final melt curve as follows: 5 min incubation at 72 °C, 15 sec at 95 °C, 15 sec at 55 °C and 15 sec at 95 °C. RT-qPCR data were analyzed with LinRegPCR analysis software. The geometric mean of the reference genes RPL3A and β-actin was used for normalization. Mouse mRNA primers used were purchased from Biolegio and the sequences are listed in Appendix A.

### 4.7. Enzyme-Linked Immunosorbent Assay (ELISA) for IL-6 and TNF-α

ELISA was performed to determine the levels of IL-6 and TNF-α in the cell supernatant collected from LRRK2 parental and KO RAW 264.7 cells seeded in a 96-well plate (20,000 cell/well) after LPS treatment for 6 and 24 h. Cell supernatants were collected and stored at −80 °C. IL-6 was detected using Mouse IL-6 DuoSet ELISA (R&D systems, DY406-05) and TNF-α was detected using Mouse TNF-α DuoSet ELISA (R&D systems, DY410-05) according to the manufacturer’s protocols. The samples were diluted 30 times before use. Absorbance was measured at 450 nm and 570 nm for background correction using a Synergy™H1 Hybrid Multi-Mode Reader (BioTek^®^ Instruments GmbH, Bad Friedrichshall, Germany). Inflammatory cytokine levels were determined using the standard samples provided in the kit.

### 4.8. Lactate Measurement

Lactate measurements were performed on the cell supernatant of LRRK2 parental and KO RAW 264.7 cells seeded in 96-well plate (20,000 cell/well) after LPS treatment for 6 and 24 h. Cell supernatants were diluted 20 times in demineralized water. A lactate standard curve for concentrations from 0–1.2 mM sodium L-lactate (Sigma, L7022-5G) was performed in each experiment to quantify the quantity of lactate produced in each sample. Lactate measurement was performed in a 96-well plate. Each well contained a mixture of 20 μL diluted supernatant or lactate standard and 225 μL reaction mixture (0.44 M glycine, 0.38 M hydrazine (pH 9.0) and 2.8 mM NAD+). Background absorbance of the plate was performed at 340 nm using the Synergy H4 for 3 min. After that, lactate dehydrogenase (LDH) (5 units) was added to each well followed by absorbance determination at 340 nm for 1 h. Lactate concentration in each sample was determined from the linear regression of the standard curve [77].

## Figures and Tables

**Figure 1 ijms-24-01644-f001:**
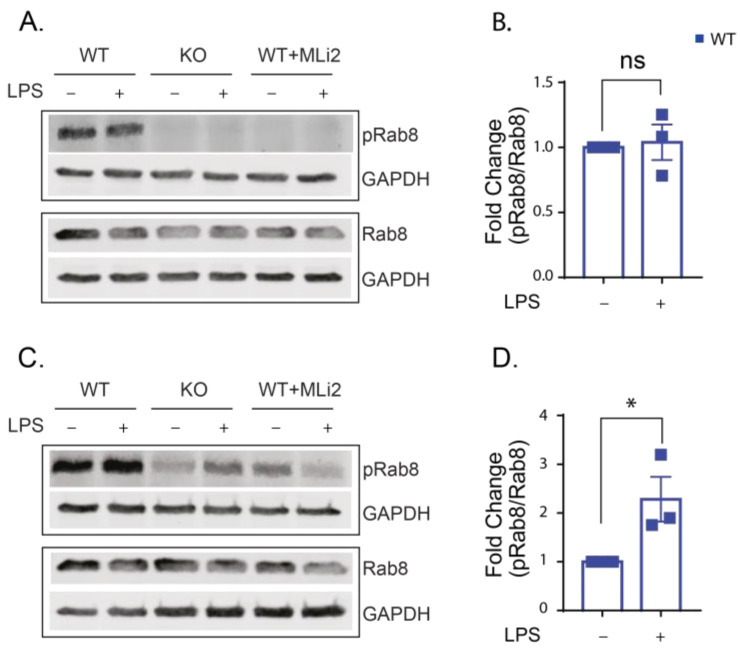
LPS stimulation for 24 h increases LRRK2 kinase activity. (**A**,**C**) Representative Western blots for lysates from WT-, KO- and MLi2-treated WT cells with and without LPS stimulation for 6 (**A**) and 24 (**C**) hours and immunoblotted with Rab8, phospho Rab8 and glyceraldehyde-3-phosphate dehydrogenase (GAPDH) antibodies (GAPDH is used as a loading control). LRRK2 kinase activity was calculated from the ratio of phospho Rab8/Rab8. (**B**) Fold change in phospho Rab8/Rab8 relative to the WT was calculated for 6 h LPS stimulation and (**D**) 24 h LPS stimulation. The experiment was performed 3 times and t-test was used to compare the ratio of phospho Rab8/Rab8 relative to the WT. Error bar represents mean ± SEM. *p*-values indicating statistically significant differences between the mean values are defined as follows: ns—not significant, * *p* < 0.05.

**Figure 2 ijms-24-01644-f002:**
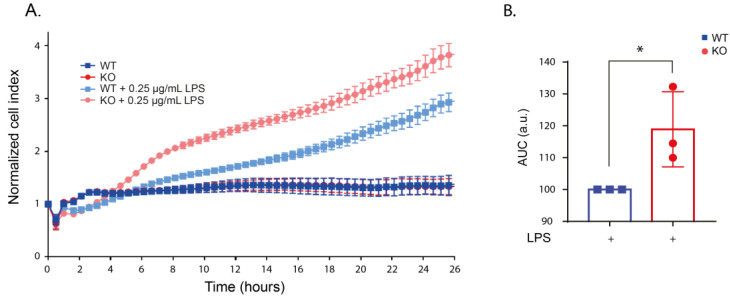
LPS stimulation altered the morphology of KO cells. (**A**) Representative real-time impedance measurement of the WT and KO cells with and without 250 ng/mL LPS stimulation using xCELLigence system. (**B**) Bar graph showing the area under curve (AUC) of 3 independent xCELLigence measurements normalized to the LPS-stimulated WT cells. t-test was used to deduce significant differences. Error bar represents mean ± SD. Significance is defined as: * *p* < 0.05.

**Figure 3 ijms-24-01644-f003:**
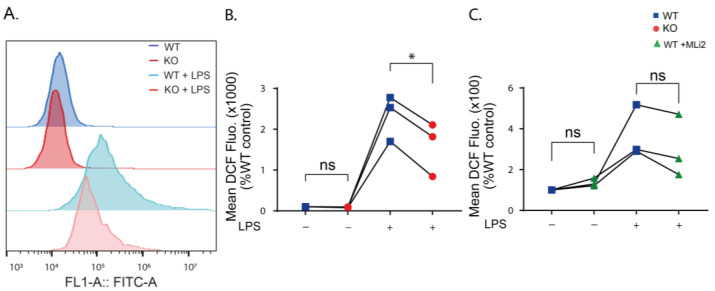
Cellular ROS levels are reduced when LRRK2 is absent or its kinase function is inhibited. (**A**) Histogram showing representative FACS sample analysis of WT and KO cells with and without LPS stimulation for 24 h using DCFDA. (**B**) Analysis of the mean DCF fluorescence in the KO cells without and with LPS stimulation normalized to the control WT. (**C**) Analysis of the mean DCF fluorescence in the MLi2-treated WT cells without and with LPS stimulation normalized to the control WT. The experiment was performed 3 times and one-way ANOVA was used to compare the mean DCF fluorescence relative to the control WT. *p*-values indicating statistically significant differences between the mean values are defined as follows: ns—not significant, * *p* < 0.05.

**Figure 4 ijms-24-01644-f004:**
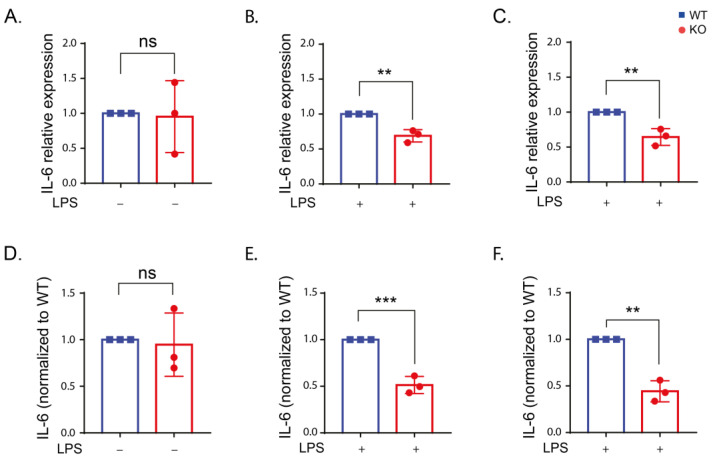
IL-6 expression is reduced in the absence of LRRK2 after LPS stimulation. (**A**–**C**) IL-6 expression in the WT and KO cells under control conditions (**A**) and after LPS stimulation for 6 (**B**) and 24 h (**C**) using reverse transcription quantitative real-time PCR using β-actin and ribosomal protein L13A (RPL13A) as housekeeping genes. (**D**–**F**) The amount of IL-6 in the supernatant of the WT and KO cells under control conditions (**D**) and after LPS stimulation for 6 (**E**) and 24 h (**F**) using ELISA assay. The experiment was performed 3 times and t-test was used to compare the average IL-6 expression or level normalized to the corresponding WT. Error bar represents mean ± SD. *p*-values indicating statistically significant differences between the mean values are defined as follows: ns—not significant, ** *p* < 0.01, *** *p* < 0.001.

**Figure 5 ijms-24-01644-f005:**
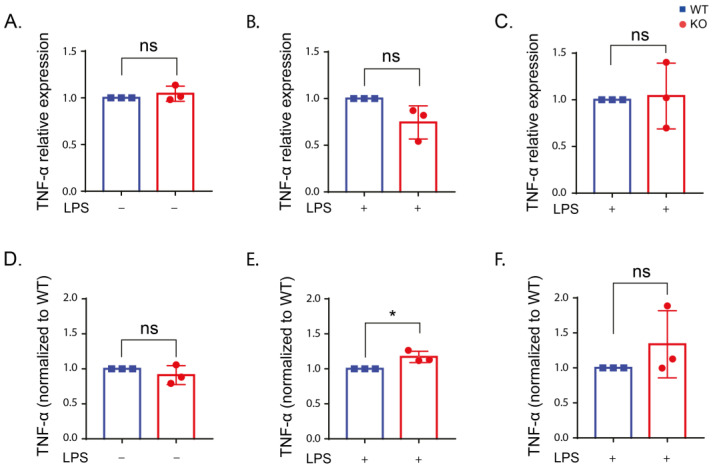
TNF-α expression is not altered in the presence or absence of LRRK2 after LPS stimulation. (**A**–**C**) TNF-α expression in the WT and KO cells under control conditions (**A**) and after LPS stimulation for 6 (**B**) and 24 h (**C**) using reverse transcription quantitative real-time PCR using β-actin and RPL13A as housekeeping genes. (**D**–**F**) The amount of TNF-α in the supernatant of the WT and KO cells under control conditions (**D**) and after LPS stimulation for 6 (**E**) and 24 h (**F**) using ELISA assay. The experiment was performed 3 times and t-test was used to compare the average TNF-α expression or level normalized to the corresponding WT. Error bar represents mean ± SD. *p*-values indicating statistically significant differences between the mean values are defined as follows: ns—not significant, * *p* < 0.05.

**Figure 6 ijms-24-01644-f006:**
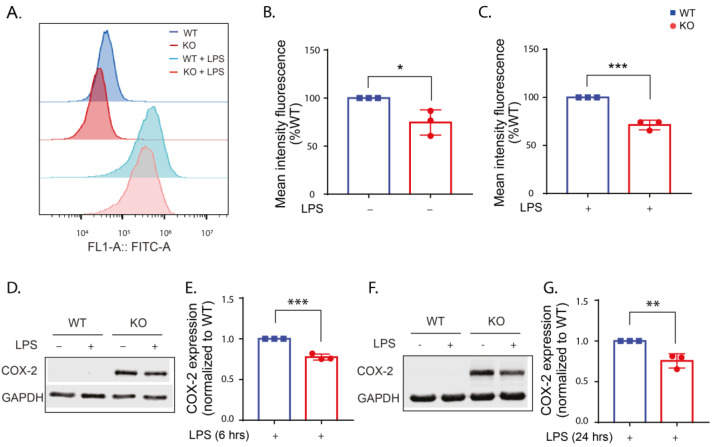
COX-2 is reduced in the KO cells with and without LPS stimulation compared to the WT. (**A**) Histogram showing representative FACS sample analysis of WT and KO cells with/without LPS stimulation for 24 h. (**B**,**C**) Analysis of the mean FITC fluorescence (representing COX-2 expression) in the WT and KO cells normalized to the corresponding WT. (**D**,**F**) Representative Western blots for lysates from WT and KO cells with and without LPS stimulation for 6 (**D**) and 24 (**F**) hours and immunoblotted with COX-2 antibody and GAPDH antibody as a loading control. (**E**) Fold change in COX-2 expression in the KO cells normalized to the WT calculated for 6 h LPS stimulation and (**G**) 24 h LPS stimulation. The experiments were performed 3 times and *t*-test was used to compare COX-2 expression in the KO cells normalized to the WT. Error bar represents mean ± SD. *p*-values indicating statistically significant differences between the mean values are defined as follows: * *p* < 0.05, ** *p* < 0.01, *** *p* < 0.001.

**Figure 7 ijms-24-01644-f007:**
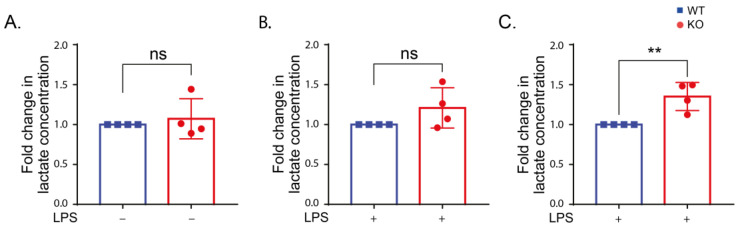
LRRK2 affects lactate production after 24 h LPS stimulation. (**A**–**C**) Lactate levels in the supernatant of WT and KO cells under control conditions (**A**) and after LPS stimulation for 6 (**B**) and 24 h (**C**). The experiment was performed 4 times and t-test was used to compare the average amount of lactate in the KO normalized to the corresponding WT. Error bar represents mean ± SD. *p*-values indicating statistically significant differences between the mean values are defined as follows: ns—not significant, ** *p* < 0.01.

**Figure 8 ijms-24-01644-f008:**
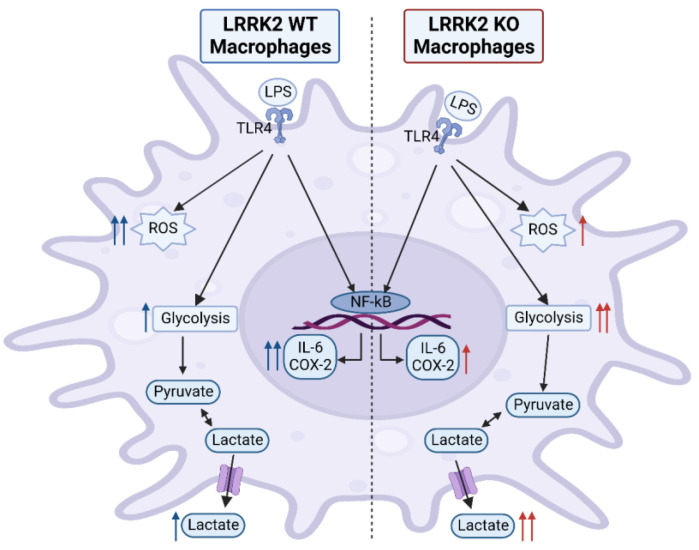
LPS stimulation effects in LRRK2 WT and KO RAW 264.7 cells. Stimulating RAW macrophages with LPS results in increased reactive oxygen species (ROS), inflammatory cytokine and lactate production. LRRK2 modulates these effects. Compared to KO cells, LRRK2 showed an increase in ROS production and IL-6 and COX-2 levels, while glycolysis was reduced, indicated by a lowered lactate level. This figure was created with BioRender.com.

## Data Availability

Not applicable.

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
