# Peer review of "Characterization of Lipopolysaccharide Effects on LRRK2 Signaling in RAW Macrophages"

_ijms, 2023, doi:10.3390/ijms24021644_

Round 1

Reviewer 1 Report

The manuscript by Oun et al. entitled: Characterization of lipopolysaccharide effects on LRRK2 signaling in RAW macrophages, uses a macrophage-like, Abelson leukemia virus-transformed cell line derived from BALB/c mice and a correspindig LRRK2 Knockout RAW line as a model to interrogate LRRK2 immune function with and without LPS stimulation for various time points.

The rationale for the study was that the literature shows inconclusive and contradictory results for LRRK2 function in immune cells. While the study does not show critical new findings of LRRK2 immune function, the authors focused on establishing phenotypes that had variable results in other model systems or experimental paradigms in ROS levels, reduction of IL-6, COX-2 expression and lactate production after LPS stimulation.

More specifically, LPS stimulation increased LRRK2 kinase activity, +LPS: KO increased cell impedance by xCELLigene suggesting alteration in cell morphology, LPS increased ROS in WT and KO, but WT higher compared to KO and kinase inhibition. In LRRK2-KO there was a reduction in IL-6, COX-2, increase in lactate

The authors provide a comprehensive discussion with detailed literature examples on the commonalities and differences to other studies with reasoning, e.g. concentration and timing of LPS stimulation.

The

Minor changes:

Please include two additional citations on inflammatory cytokines in PD patients, since such studies are scarce and difficult to perform, but critical to develop fluid biomarkers.

Brockmann et al. 2016, https://doi.org/10.1186/s12974-016-0588-5

Brockmann et al. 2017, https://doi.org/10.1111/ene.13223

Minor typos for special charactors, probably due to conversion of document.

The article is in line with the IJMS scope and should be accepted for publication after minor revision. Results and Conclusions are in clear English language to the reader and easy to follow.

Author Response

Dear reviewer 1,

Please find enclosed the response to your comments. We thank you for your effort and valuable comments.

Reviewer 2 Report

Review Oun et al,

The authors explored the role of LRRK2 in the immune system using RAW macrophages LRRK2 KO and compared to control. They assessed i) LRRK2-substrate phosphorylation (Rab8), ii) cell morphology, iii) ROS and inflammatory cytokine levels and iv) lactate production upon LPS treatment. Overall, the authors concluded that LRRK2 depletion causes impaired response to inflammatory stimuli.

The manuscript is well written and organised. However, my main comment is related to quantification and statistic. T test is not adequate here. The graphs (from fig 1 to fig.7) should include all the conditions and the appropriated statistical test should be applied.

Why the error bar is not present in one of the conditions?

Figure 1: it is very difficult to read, what light colours stand for? The authors should show one graph (genotype and tretament) per blot and apply the adequate statistical test.

Western blot seems overexposed.

1C: how the authors explain the P Rab8 in the KO condition?

Evaluation of the LRRK2 level should be add.

Figure 2: how the authors exclude that KO cells are dividing quickly?

Figure 3: again, the comparison here should be TWO-WAY ANOVA and one single graph. See comments in Figure1

Figure 6: Which is the ‘housekeeping gene’ used for normalisation in the FACS experiment? The authors should perform western blot as an independent experiment.

Author Response

Dear reviewer 2,

Please find enclosed the response to your comments. We thank you for your effort and valuable comments.

Reviewer 3 Report

Fig 3 Implies that LRRK2 modulates IL-6 level. However, this figure does not include the effect LRRK2 inhibitor has on IL-6 levels. Given LRRK2 inhibitors' importance as a potential therapeutic strategy and validate the overall implication of this figure, the authors should include in this figure if inhibiting LRRK2 kinase activity is as effective as the KO results. 

Author Response

Dear reviewer 3,

Please find enclosed the response to your comments. We thank you for your effort and valuable comments.

Round 2

Reviewer 2 Report

The authors improved the manuscript. However, the statistical representation (t-test) is still not adequate.
